# Assessing the Potential Impact of a Long-Acting Skin Disinfectant in the Prevention of Methicillin-Resistant *Staphylococcus aureus* Transmission

**DOI:** 10.3390/ijerph17051500

**Published:** 2020-02-26

**Authors:** Christopher T. Short, Matthew S. Mietchen, Eric T. Lofgren

**Affiliations:** Paul G. Allen School for Global Animal Health, Washington State University, Pullman, WA 99164, USA; christopher.short@wsu.edu (C.T.S.); matthew.mietchen@wsu.edu (M.S.M.)

**Keywords:** MRSA, decolonization, hospital epidemiology

## Abstract

Healthcare-associated transmission of methicillin-resistant *Staphylococcus aureus* (MRSA) remains a persistent problem. The use of chlorhexidine gluconate (CHG) as a means of decolonizing patients, either through targeted decolonization or daily bathing, is frequently used to supplement other interventions. We explore the potential of a long-acting disinfectant with a persistent effect, immediate decolonizing action in the prevention of MRSA acquisition, and clinical illness and mortality in an 18-bed intensive care unit, based on a previous model. A scenario with no intervention is compared to CHG bathing, which decolonizes patients but provides no additional protection, and a hypothetical treatment that both decolonizes them and provides protection from subsequent colonization. The duration and effectiveness of this protection is varied to fully explore the potential utility of such a treatment. Increasing the effectiveness of the decolonizing agent reduces colonization, with a 10% increase resulting in a colonization rate ratio (RR) of 0.89 (95% CI: 0.89,0.90). Increasing the duration of protection results in a much more modest reduction, with a 12-hour increase in protection resulting in an RR of 0.99 (95% CI: 0.99, 0.99). There is little evidence of synergy between the two.

## 1. Introduction

The prevention of healthcare-associated infections (HAI) is a pressing issue to the medical community, governmental agencies, and patient advocacy groups. While there are relatively straightforward interventions, such as hand hygiene, that can deliver substantial gains, some pathogens such as methicillin-resistant *Staphylococcus aureus* have proven stubbornly persistent, causing an estimated 120,000 bloodstream infections and 20,000 deaths in the United States [1]. This necessitates the use of the “Swiss Cheese” approach, where multiple imperfect interventions (due to non-compliance, clinical demands, antimicrobial resistance, etc.) are put in place in the hope that, when combined, they will have a substantial effect on reducing the burden of HAIs.

The use of chlorhexidine gluconate (CHG) is one such intervention, intended to disinfect the skin of a patient, and is applied either immediately before a procedure (such as surgery) or as part of a daily bathing regimen. A meta-analysis by Kim et al. [2] suggests that the use of CHG bathing is effective; though some community studies have shown less compelling results [3]. Previous modeling work by Lofgren et al. [4] estimates that the actual per-use effectiveness of CHG bathing is relatively low, suggesting the potential for substantial improvement in either the administration or mechanism of action. 

One such potential intervention is increasing the duration of the protective effect of a decolonizing agent, providing not only immediate disinfection/decolonization, but also persistent protection against colonization from later hand hygiene failures or environmental exposure. Mathematical modeling is an ideal tool to explore the potential impact of a hypothetical intervention, examining under what conditions they might be effective and providing important quantitative information to drive future empirical studies [5,6]. In this paper, we explore the potential for a hypothetical CHG-like disinfectant, exploring the impact of a persistent protective effect by varying effectiveness and duration using a mathematical model.

## 2. Materials and Methods 

### 2.1. MRSA Transmission Model

Extending the previous model of an intensive care unit (ICU) [7,8], including model-based estimates of CHG effectiveness [4], we represented the transmission of MRSA as a stochastic compartmental model, which captures both the current colonization status of a patient, as well as the presence or absence of MRSA on the hands of clothing of both nurses and physicians. Nurses and physicians can be uncontaminated (N_U_ and D_u_ respectively) or contaminated (N_C_ and D_C_ respectively), and may transition between these states by coming into contact with a colonized patient (moving from uncontaminated to contaminated) or by washing their hands or changing their personal protective equipment (typically gowns and gloves) which moves them in the opposite direction. Patients are modeled as being presently colonized with MRSA (P_C_), uncolonized (P_U_), or protected (P_P_), denoting those patients who are both uncolonized and not at risk for colonization. Colonized patients may progress to a further category representing those with MRSA-related bacteremia (P_B_) (Figure 1).

The 18-bed ICU was broken down into six groups consisting of three patients each occupying a single room and a single assigned nurse. A single dedicated intensivist was assumed to see all patients. The ICU itself was assumed to be at full and constant occupancy (i.e. a discharged patient was immediately replaced), providing a steady population. While this highly structured population is a simplification of the realities of an intensive care unit, previous work [7] has shown this to be a more conservative assumption when estimating the impact of a given intervention when compared to a less structured population where all nurses are assumed to care for all patients through random mixing (Figure 2).

As with all models, this model makes several simplifying assumptions. While other types of healthcare workers (HCWs) are employed by hospitals, they are not represented within this model. As patients are assumed to be immobile and thus cannot interact with each other, all transmission must be mediated by a healthcare worker in some form. Additionally, patients are assumed to be identical in terms of the difficulty of caring for them (and thus the rate at which HCWs interact with them), their risk of MRSA acquisition and subsequent negative outcomes, etc. The hospital itself is assumed to follow Centers for Disease Control and Prevention (CDC) contact precaution guidelines, putting MRSA colonized patients under contact precautions with perfect and instant accuracy. Further, all HCWs are assumed to wash their hands after each interaction with either the patient or their direct care environment (hereafter referred to as a direct care task), following the Five Moments of Hand Hygiene put forward by the World Health Organization (https://www.who.int/infection-prevention/campaigns/clean-hands/5moments/en/), and to do so regardless of whether or not they wear personal protective equipment such as gloves or gowns as part of a contact precautions protocol. These assumptions are an attempt to represent the environment of a well-equipped and well-staffed hospital with no major outstanding failings in their infection control program. By and large, the model used parameter values from previous models and a full description of the model system may be found in [7]. The interpretation of each parameter in the model, its value (or distribution), and the source it was drawn from may be found in Table 1.

### 2.2. Decolonization Intervention

Decolonization was modeled as the periodic application of a CHG-like compound that moved the patient from either the P_U_ or P_C_ compartments to the P_P_ compartment, representing both immediate MRSA decolonization (if present) as well as a period of protection wherein the patient was effectively immune from MRSA colonization. This protection eventually wanes, moving the patient back to the P_U_ compartment. In order to explore the parameter space for the hypothetical CHG-like compound’s effectiveness and duration—as well as any interaction between the two—these parameters were varied randomly (see Table 1). While the simulated effectiveness of the compound varies continuously between 0.0 and 0.45, we present the results with an interval of 0.10, representing a 10% change—a substantial, but feasible improvement. The default application rate for the disinfectant was assumed to be once daily at 24-hour intervals.

### 2.3. Stochastic Simulation

The acquisition of MRSA, as well as subsequent cases of MRSA-related bacteremia and death were stochastically simulated using Gillespie’s Direct Method [12] (using the StochPy package [13] in Python 3.2. The ICU was simulated for one year, with the cumulative incident acquisitions, bacteremia cases and MRSA-related deaths recorded as outcome variables for 10,000 iterations of the model. The impact of the hypothetical intervention was assessed using a Poisson regression model with terms for the effectiveness of the agent, the duration of protection, and the interaction of the two using standard *glm* functions in R. Scatter plots of the simulations were created using MATLAB. The code and results used in this manuscript are available at http://www.github.com/epimodels/protective_decol. 

### 2.4. Ethics

As this is a mathematical modeling study using previously published data and no human subjects, it does not require the approval of an Institutional Review Board.

## 3. Results

The results of the stochastic simulations show that the effectiveness of the hypothetical disinfecting agent—regardless of whether or not there is any subsequent protective effect from re-colonization—has the greatest impact on incident MRSA acquisitions (Figure 3). Numerically, a 10% increase in application effectiveness results in a colonization rate ratio (RR) of 0.89 (95% Confidence Interval (CI): 0.89, 0.90), corresponding to an almost 1:1 relationship between improved decolonization effectiveness and the resulting drop in acquisitions. This same pattern can be seen for cases of MRSA bacteremia (Figure 4), where a 10% increase in application effectiveness results in a RR of 0.76 (95% CI: 0.75, 0.78). The simulations suggested little to no impact for bacteremia-related deaths, with an RR of 0.99 (95% CI: 0.97, 1.00).

The improvements seen with the addition of a long-lasting protective effect were much more modest. A 12-hour increase in protection resulted in an RR of 0.99 (95% CI: 0.99, 0.99), with no evidence of an impact on incident bacteremia or MRSA-related deaths, which both had RRs of 1.00 (95% CI: 0.99, 1.01). Additionally, there was no evidence of clinically meaningful interaction, either synergistic or antagonistic, for any of the three outcomes.

## 4. Discussion

The results clearly indicate that improvements to the effectiveness of skin decolonization protocols, which Lofgren et al. estimate are currently well below 50% [4], are likely to reap considerable benefits in terms of reducing overall MRSA acquisitions and subsequent clinical illness and death. Comparatively, the introduction of a hypothetical long-acting protective effect from these agents is rather modest in its effect. While the two are clearly not mutually exclusive, it is important to be mindful of potential tradeoffs. For example, a hypothetical compound would have to have a staggeringly long protective effect in order to offset even a small decrease in effectiveness. Such a decrease could be the result of the compound itself, such as increased skin irritation resulting in skipped applications, or could be the result of increased difficulty in application.

Even without tradeoffs, these results suggest, even under nearly perfect study conditions, that the expectations for the effect of any long-acting skin disinfecting agent should be modest and that randomized trials seeking to measure such an effect should be powered accordingly. This is further highlighted by the attenuation of the results for bacteremia and MRSA-related mortality. These clinical endpoints are further down the causal chain from the intervention, which specifically targets the initial acquisition of MRSA. Many further steps are required to develop a clinical case of MRSA-related bacteremia, and from there, death is, thankfully, a relatively rare outcome. As such, these outcomes are subject to considerably more stochasticity, and the effect of the intervention (if any) is lost to the “noisier” process. Thus, while it may be tempting to use these outcomes as they are easier to collect using administrative data and are often the focus of patient advocacy groups, they may be difficult if not impossible to use successfully. Recent work by Blanco et al. [14] highlights this difficulty for a number of HAI-related outcomes.

## 5. Conclusions

The development of long-acting disinfecting and sterilizing agents is of considerable interest in hospital epidemiology, both to improve environmental disinfection and also to provide long-lasting decolonization for patients. While the former is unexplored in this paper, our results suggest that the latter may be, at best, only modestly effective at decreasing incident acquisitions of MRSA. In contrast, improvements to decolonizing agents that improve effectiveness—either new compounds or improvements to the administration of existing compounds—seem likely to result in substantial dividends. Previous work by Lofgren et al. estimating the per-application effectiveness of CHG based on existing RCTs suggests there is considerable room for such improvements.

While this model is rooted in a hypothetical intervention, these results may still have practical application. The decision to invest the limited resources available to hospital epidemiology as a field that invariably necessitates some degree of economic tradeoffs. The limited impact of increasing the duration of a decolonizing compound’s protective effect suggests that this is not a high-priority area of basic research. Additionally, the model’s estimates may be used in the design of randomized trials of future decolonizing compounds—including power calculations—and they demonstrate the difficulty of definitively showing the impact of a new intervention with only a small effect given the realities of infection control, which often involve small sample sizes and highly stochastic outcomes.

As with all modeling studies, the necessary simplification of real-world interactions may result in errors. While every attempt has been made to use a model that captures the salient features of the within-hospital MRSA transmission process (while remaining tractable), it is possible that errors may have arisen from simplifying assumptions, parameters drawn from populations with different underlying unmeasured covariates, or other sources of model error. The authors tried to minimize these by drawing from estimates largely from academic medical settings and conducting an extensive sensitivity analysis of the basic model form (see [7]). Despite this, in the evaluation of hypothetical interventions, mathematical models remain a powerful tool for providing a quantitative, structured examination of the expected effects in a framework that can be critiqued, extended, and modified. This work is not intended as the final word on long-acting skin disinfectants, but rather as a work exploring their potential, guiding future clinical studies, and properly calibrating expectations as to their effect on patient care.

## Figures and Tables

**Figure 1 ijerph-17-01500-f001:**
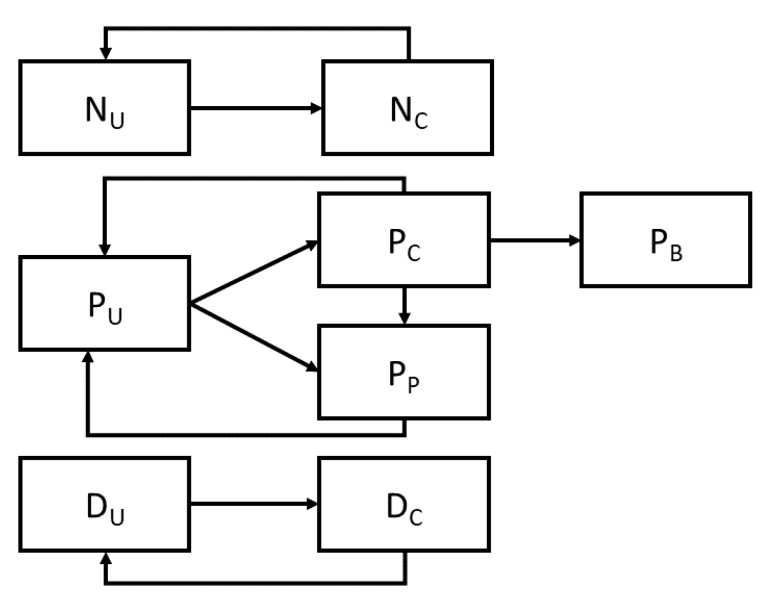
Schematic representation of the compartmental flow of a mathematical model of methicillin-resistant *Staphylococcus aureus* transmission in an intensive care unit. Arrows indicate possible transition states. Nurses and doctors are classified as uncontaminated or contaminated (N_U_/D_U_ and N_C_/D_C_ respectively), while patients may be uncolonized (P_U_), colonized (P_C_), temporarily protected from colonization (P_P_) and bacteremic (P_B_).

**Figure 2 ijerph-17-01500-f002:**
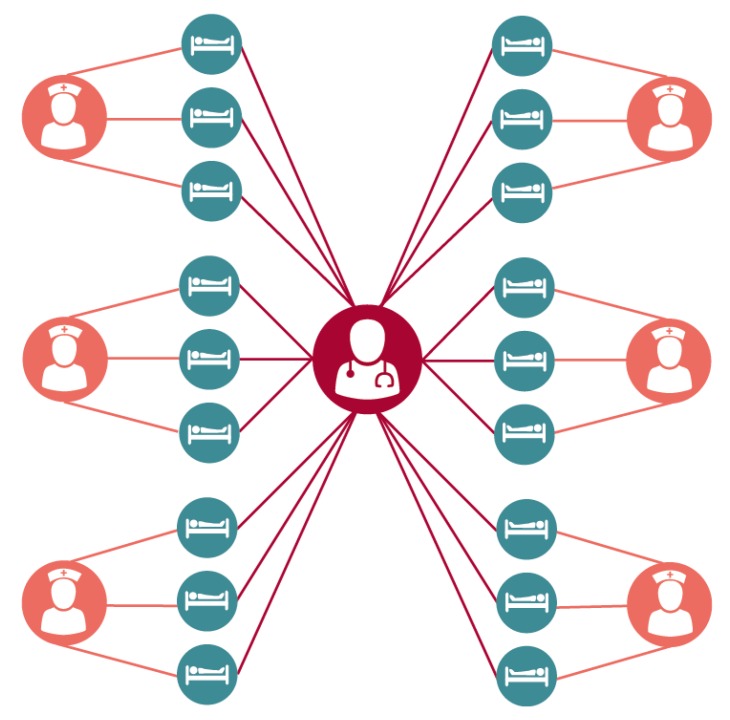
Structured layout of an intensive care unit. Patients (teal) are treated by a single assigned nurse (orange), while a single intensivist (red) randomly treats all patients in the ward. Figure by Eric Lofgren is licensed under CC-BY-4.0.

**Figure 3 ijerph-17-01500-f003:**
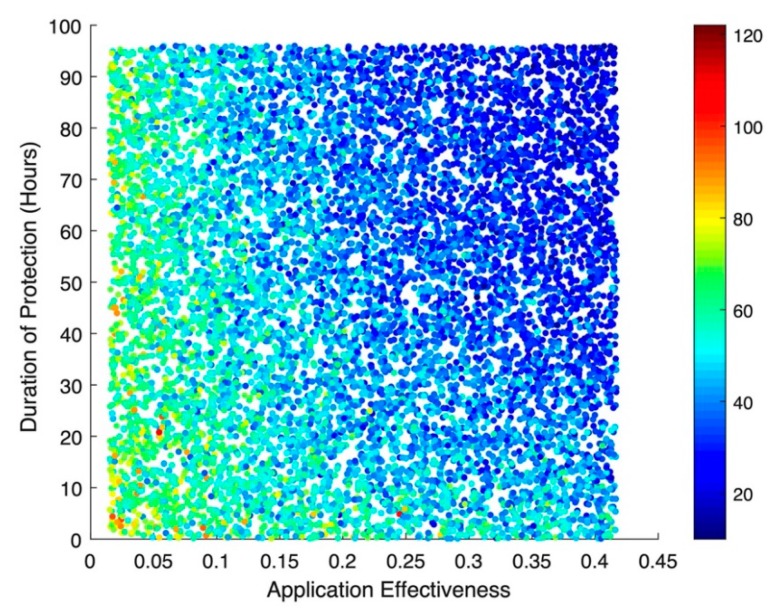
Scatterplot of incident MRSA acquisitions in an 18-bed ICU over a simulated year. Each dot represents one of 10,000 iterations of a stochastic model, simulating a spread of possible levels of effectiveness and duration of protection for a hypothetical long-acting decolonization agent.

**Figure 4 ijerph-17-01500-f004:**
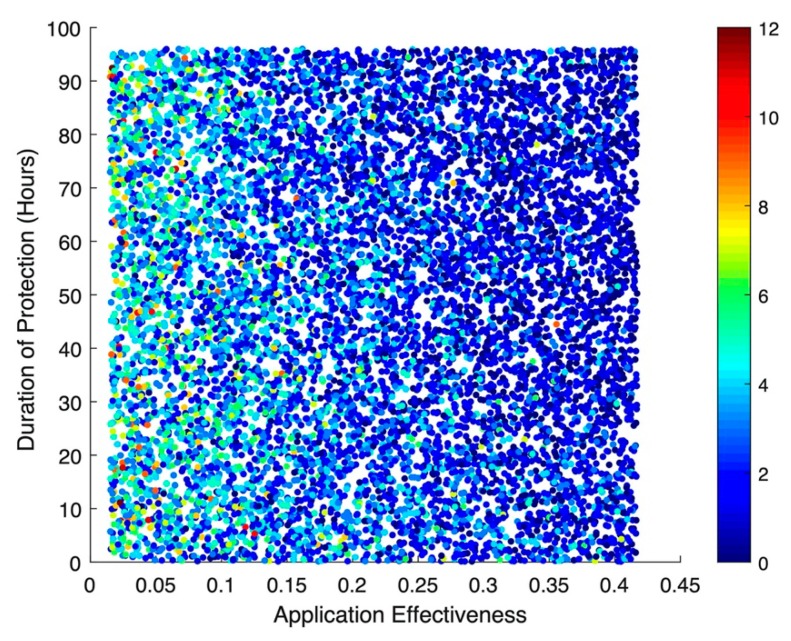
Scatterplot of incident MRSA-related bacteremia cases in an 18-bed ICU over a simulated year. Each dot represents one of 10,000 iterations of a stochastic model, simulating a spread of possible levels of effectiveness and duration of protection for a hypothetical long-acting decolonization agent.

**Table 1 ijerph-17-01500-t001:** Parameter values for a model of methicillin-resistant *Staphylococcus aureus* (MRSA) transmission within an ICU.

Parameter	Value ^1^	Interpretation
$\rho_{N}$	3.973	Contact rate between patients and nurses (direct care tasks/hour)
$\rho_{D}$	0.181	Contact rate between patients and physicians (direct care tasks/hour)
σ	0.054	Probability of hand contamination on contact with colonized patient
ψ	0.0527	Probability of colonization given contact with contaminated healthcare workers (HCW)
θ	0.00949	Probability of discharge from the ICU
ν	0.0779	Proportion of admissions colonized with MRSA
$\iota_{N}$	6.404	Effective nursing hand decontamination rate assuming 56.55% compliance and 10.682 direct care tasks per hour
$\iota_{D}$	1.748	Effective physician hand decontamination rate assuming 56.55% compliance and 3.253 direct care tasks per hour
$\tau_{N}$	2.728	Nurse gown and glove changes per hour assuming 82.66% compliance
$\tau_{D}$	0.744	Physician gown and glove changes per hour assuming 82.66% compliance
μ	0.002083	Natural decolonization rate
$\epsilon_{E}$	U(0.15,0.417)	Effectiveness of chlorhexidine gluconate (CHG)-like compound
$\epsilon_{U}$	U(0.1,96)	Duration of protection of CHG-like compound (hours)
$\epsilon_{B}$	0.04167	Decolonization application frequency (24 hours^−1^)
$\delta_{I}$	0.000343 [9]	Rate colonized patients develop bacteremia
$\delta_{D}$	0.000403 [10]	Rate at which bacteremic patients die
$\delta_{C}$	0.002976 [11]	Rate at which bacteremic patients recover
$\delta_{P}$	0.32 [9]	Proportion of colonized admissions that have bacteremia

^1^ Unless otherwise noted, the source for parameter values may be found in [7].

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
