# Peer review of "Assessing the Potential Impact of a Long-Acting Skin Disinfectant in the Prevention of Methicillin-Resistant Staphylococcus aureus Transmission"

_ijerph, 2020, doi:10.3390/ijerph17051500_

Round 1

Reviewer 1 Report

since the paper only refers to 2 papers with more detailed descriptions it would useful to expand the model description in this paper by integrating some details from thes previlous papers eg, Figure 2 in the 2019 is more useful than Figure 1; use that instead of the current Figure 1 Figure 3 in the 2019 paper illustrates nicely the setup of the model; include it here Parameter description in Table 1 of the 2019 is better than the one in the present manuscript; use the 2019 Table 1 line 62: PI should be PB Figure 1: right top compartment should be labeled "NC" line 113: link to repository does not work; please fix I was not able to look at he actual code Results section is rather short why was 10% increase chosen? what about 5, 15, 20? this is hinted at in lines 101-103 but results are not presented line 139: sentence is rather long; consider breaking it up

Author Response

Response to Reviewer 1:

since the paper only refers to 2 papers with more detailed descriptions it would useful to expand the model description in this paper by integrating some details from thes previlous papers eg, Figure 2 in the 2019 is more useful than Figure 1; use that instead of the current Figure 1 Figure 3 in the 2019 paper illustrates nicely the setup of the model;

We have elected to keep Figure 1 in this paper – we believe that it is a better representation of the model, and with the considerably more complex patient care pathways and potential outcomes of the paper, the Figure 2 in the 2019 paper would give the reader a false sense of the structure of the model.

Figure 3 in the 2019 paper has been added here as Figure 2 with the appropriate license.

 include it here Parameter description in Table 1 of the 2019 is better than the one in the present manuscript; use the 2019 Table 1

The parameter descriptions in Table 1 have been considerably reworked. While not duplicative with the 2019 paper, they should be considerably more descriptive here.

 line 62: PI should be PB

This has been corrected.

Figure 1: right top compartment should be labeled "NC"

This has been corrected.

line 113: link to repository does not work; please fix I was not able to look at he actual code

The repository in question was errantly left “Private” and not available for public viewing. It has now been made public and should be visible.

Results section is rather short why was 10% increase chosen? what about 5, 15, 20? this is hinted at in lines 101-103 but results are not presented line

We admit that the choice of 10% is somewhat arbitrary. Essentially, it was chosen to provide visible separation between the three outcome rate ratios, which at a single percentage change are much harder to see as distinct. 10% seemed a reasonable threshold for a substantial, but not unthinkable, improvement in the disinfecting quality of the hypothetical agent. Some discussion of this has been added to the methods section.

As for the length of the results section, that largely emerges from the targeted nature of the study in question. We feel that the results, as presented, represent sufficient information to make an informed judgement in this area, without containing unnecessary padding.

139: sentence is rather long; consider breaking it up

This sentence has been restructured.

Reviewer 2 Report

Thank you for doing this important work. Given the high rates of hospital-acquired infections globally, this is an important topic. I hope that the following points will help you to strengthen your work before publication.

Abstract

The findings report rates percentages. Please correct the presentation of CI by removing comma symbols and inserting spaces on either side of the hyphens. For example, 95% CI = 0.89 – 0.90) in all text

Materials and Methods

Specify what type of study was carried out

The authors include doctors and nurses in the sample. In some countries nursing does not take care of the patient's toilet, there is another status. Please clarify.

Line 80. What are HCWs? I suppose that the authors mean health care workers. Please indicate the meaning of the abbreviations for the first time

Line 86. Absence of reference for “Five Moments of Hand Hygiene put forward by the World Health Organization”

References: Review bibliographic references number there are many mistakes

Author Response

The findings report rates percentages. Please correct the presentation of CI by removing comma symbols and inserting spaces on either side of the hyphens. For example, 95% CI = 0.89 – 0.90) in all text

We can find no reference to how CIs should be formatted in the instructions for authors for IJERPH. However, the format we are using is standard for several other epidemiology journals (including the American Journal of Epidemiology). We have elected to leave our format here unchanged, as we believe the risk of introducing inconsistencies is greater than any improvement to readability. However, we are happy to leave this to the discretion of the editor.

Specify what type of study was carried out

A statement has been added to the end of the sentence in line 49.

The authors include doctors and nurses in the sample. In some countries nursing does not take care of the patient's toilet, there is another status. Please clarify.

This has been specified on line 82 now.

Line 80. What are HCWs? I suppose that the authors mean health care workers. Please indicate the meaning of the abbreviations for the first time

This has been defined within the text.

Line 86. Absence of reference for “Five Moments of Hand Hygiene put forward by the World Health Organization”

The Five Moments primarily exist as internet-based information rather than something more amenable to formal citation. An appropriate link has been added to the text.

References: Review bibliographic references number there are many mistakes

Significant corrections have been made to the references.

Reviewer 3 Report

The suject assesed in this paper is a big problem in hospital in general and ICU in particular. The importance od emidemiology is also stated

I do not see any Ethic commeetee recommendation or any informed consent from the participants , was it not necessary for such a study?

In the materials and methods I do not see any statistical explanation (wich methods and softwar used)

Its a very small number of patients,  although the rational is interesting it would required larger scale muticentric studies to reach any conclusion. 

Author Response

I do not see any Ethic commeetee recommendation or any informed consent from the participants , was it not necessary for such a study?

This study is a purely mathematical modeling-based study using previously published data, and as such does not require ethics approval. A note specifying this has been added.

In the materials and methods I do not see any statistical explanation (wich methods and softwar used)

These may be found in section 2.3, Stochastic Simulation. Some detail on the Poisson model’s software was added.

Its a very small number of patients,  although the rational is interesting it would required larger scale muticentric studies to reach any conclusion. 

Keep in mind that this is a mathematical modeling study, and the sample size is not a single 18-bed ICU, but 10,000 18-bed ICUs. One could think of this as a simulation of an impossibly large multicenter study.

Reviewer 4 Report

The intensive care unit patients are susceptible to a number of common health-care infection including MRSA strains. And therefore also great care must be taken to curb and eliminate the chances of disease transmission. The presented project on conditions which can be effective and provide persistent protective effect in ICU are very interesting. This mathematical modeling is for me personally new idea but for the future will be useful for the most severe sick patients. I do believe you will continue presented studies and probably in the future will able to prepare effective program to to protect patients in ICU for HAI.

Author Response

The intensive care unit patients are susceptible to a number of common health-care infection including MRSA strains. And therefore also great care must be taken to curb and eliminate the chances of disease transmission. The presented project on conditions which can be effective and provide persistent protective effect in ICU are very interesting. This mathematical modeling is for me personally new idea but for the future will be useful for the most severe sick patients. I do believe you will continue presented studies and probably in the future will able to prepare effective program to to protect patients in ICU for HAI.

We thank the reviewer for their comments, and hope that this will help advance the field of infection prevention for patients in the ICU in the future.